# Label-aware distance mitigates temporal and spatial variability for clustering and visualization of single-cell gene expression data
Shaoheng Liang [1,2,3] ✉, Jinzhuang Dou[1], Ramiz Iqbal[1] & Ken Chen [1] ✉

Clustering and visualization are essential parts of single-cell gene expression data analysis. The Euclidean distance used in most distance-based methods is not optimal. The batch effect, i.e., the variability among samples gathered from different times, tissues, and patients, introduces large between-group distance and obscures the true identities of cells. To solve this problem, we introduce Label-Aware Distance (LAD), a metric using temporal/spatial locality of the batch effect to control for such factors. We validate LAD on simulated data as well as apply it to a mouse retina development dataset and a lung dataset. We also found the utility of our approach in understanding the progression of the Coronavirus Disease 2019 (COVID-19). LAD provides better cell embedding than state-of-the-art batch correction methods on longitudinal datasets. It can be used in distance-based clustering and visualization methods to combine the power of multiple samples to help make biological findings.

Gene expression reflects the identity of a cell. Single-cell RNA sequencing (scRNA-seq) technologies profile thousands of cells simultaneously[1], enabling trajectory inference to reveal the course of cell development and transformation[2]. Although a large amount of data has been gathered from different tissues among large cohorts of patients[3], technical variances among separately assayed samples often overshadow the similarity of cells, resulting in disconnected trajectories (Fig. 1a) that hinders the discovery of underlying biological processes. This phenomenon often called the batch effect, complicates the single-cell sequencing data analysis. For samples collected from the same condition (i.e., same time and tissue but different participants), differences among samples are usually considered batch effects and get corrected, but over-correction is suspected in some cases[4]. For longitudinal data, where true biological changes and nuisance factors are entwined (Figure 1b), the distinction between biological and batch effects becomes more elusive.

As a practical example, researchers have collected data from the developing retina of 13 mice at different ages[5]. Ideally, the embedding of such data should reveal a fluent trajectory of how pluripotent stem cells differentiate/evolve into multipotent stem cells, and finally to specific cell types. However, as a mouse does not survive the tissue extraction, each sample in the dataset is from a unique mouse. The batch effect exists among samples collected from different mice. Because there is no bijective (i.e., injective and surjective) mapping for cells from different samples, existing methods addressing batch effect in longitudinal data assume that the features are measured on the same set of entities (cells) at different time points[6] are not applicable to single-cell data.

To address this issue, multiple methods have been published[7]. For example, Limma fits a linear model and removes the component for the batch effects[8]. Seurat utilizes mutual nearest neighbor (MNN, in CCA or RPCA space) to identify similar clusters in different batches and integrates those batches by removing the differences[9]. Harmony also integrates proximal clusters, but in an iterative way through soft clustering[10]. Liger uses nonnegative matrix factorization (NMF) to separate common and sample-specific features[11]. A neural network approach, scVI, combines a variational autoencoder and a zero-inflated model to visualize and correct the data. Notably, Harmony generates integrated clusters and visualization without giving a corrected expression profile. It is deemed acceptable, however, because corrected datasets are often hard to interpret. Researchers usually only cluster and visualize the data using such methods, and recur to statistical tests that control for the batch effect in the downstream analyses[12]. These methods have been utilized in a few large-cohort studies and satisfyingly removed the batch effect among samples[9,10]. However, none of them are designed for longitudinal data.

[1]Department of Bioinformatics and Computational Biology, MD Anderson Cancer Center, Houston, TX, USA. [2]Department of Computer Science, Rice University, Houston, TX, USA. [3]Present address: Ray and Stephanie Lane Computational Biology Department, School of Computer Science, Carnegie Mellon University, Pittsburgh, PA, USA. ✉e-mail: shaohengliang@gmail.com; kchen3@mdanderson.org

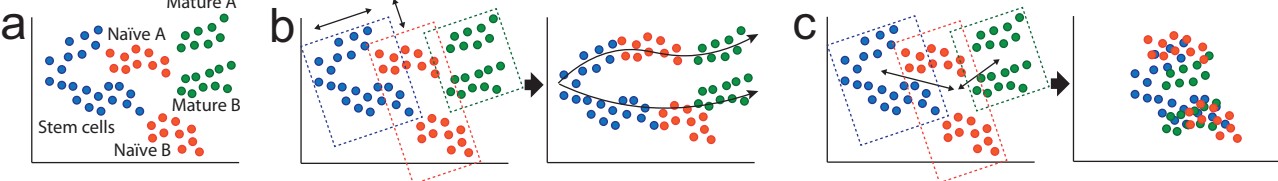

**Fig. 1 | Illustration of how LAD removes batch and reconstruct trajectories from longitudinal data. a** The raw data. **b** LAD removes true batch effects using a label-aware approach. **c** Other methods overcorrect the batches.

In an effort to address this issue, we noticed the temporal locality among the single-cell samples. Specifically, samples gathered at closer time points are expected to be more biologically alike. If we decompose the difference between two samples to batch effect and biological effect. The former is uniform between all samples, but the latter increases with the distance between samples. Thus, the relative strength of the effects decreases over distance. In other words, the observed difference between two adjacent time points is more likely to be the batch effect, than that of two distant time points.

Here, we define a distance metric, termed Label-Aware Distance (LAD) which exploits the locality to precisely remove the batch effect (Figure 1c) but keeps biologically meaningful information that forms the trajectory. As a distance metric, it is naturally compatible with all state-of-the-art distance-based clustering and embedding methods. We compare LAD with Euclidean distance, Limma, Seurat integration (with CCA and RPCA), and Harmony on simulated data and mouse retina development data, and show applications of LAD on COVID-19 patient data and human fetal lung development data. The results clearly show the benefit of LAD to biology studies.

## Results
### The label-aware distance
Across single-cell samples, genes (and their combinations) are differentially affected by the batch effect. Traditional clustering and trajectory inference methods based on metrics that assign equal weights to all genes, such as the Euclidean distance, fail to address this discrepancy in data. We introduce Label-Aware Distance (LAD), where a weight matrix (covariance matrix) for genes is used to offset the batch effect. The matrix is directly inferred from the gene expression data and time labels or spatial coordinates (Methods), based on the temporal/spatial locality intuition: the batch effect is uniform across samples, while the biological effect increases over the distance of samples.

To validate LAD, we simulated a dataset with seven samples. We assume that all cells are differentiated from an initial state, stem cells named S. S differentiates into two multipotent cell types A and B, which differentiate into terminal types A1, A2 and B1, B2, respectively (Fig. 2a). We simulated 200 genes, and assigned a unique ideal gene expression profile for each cell type, denoted as $s$, $a$, $b$, $a_1$, $a_2$, $b_1$, and $b_2$. Because the development of the cell types is gradual[13], we used weighted mean to represent the process in the simulated samples. For example, to simulate intermediate states between A and A1, we used $(\theta a + (1 - \theta)a_1)$ as its profile, where $\theta$ is set to be uniformly distributed in a range corresponds to the developing stage of a sample (Table 1). The ambiguous cell types ($\theta$ close to 0.5) are denoted as "A → A1", while cells more similar to "A" or "A1" ($\theta$ close to 0 or 1) are labelled as them each.

We randomly selected a set of genes to be affected by the batch effect and added different additive components to those genes in different samples. For every single cell, we further added random noise into the ideal profile and used a Poisson distribution to generate the gene expression. The number of samples at each time point and compositions of samples are shown in Table 1. It can be seen that the cell types gradually evolve over time.

We used Seurat to process the data with LAD, Euclidean distance, Limma, Seurat integration (with CCA and RPCA), and Harmony. In the process, 150 highly variable genes are selected, and 30 PCs are used. The result of LAD is shown in Figure 2b. Given the prior knowledge that all cell types originate from S, it can be clearly seen that it branches to A and B, and further to A1, A2, and B1, B2, respectively. Batch effect structure can still be seen in A1, A2, B1, and B2, but the downstream visualization is improved as samples are now

organized by their time. Day 1 is at the center, day 4 is separated into A1 and A2 at the top-left and B1 and B2 at the bottom-right corner, while days 2–3 correspond to the trajectory of which the cells differentiate.

In contrast, Euclidean distance (on PCA), the baseline, does not correct for any batch effect. The difference between Euclidean distance and LAD is apparent when visualizing the similarity of cell types (Supplementary Fig. 1 and Supplementary Note 1). Consequently, the terminal types A1, A2, B1, and B2 each splits into two groups, corresponding to different samples. It also fails to illustrate the evolution of the cell types. Limma does not generate a visually appealing figure, though a reasonable integration is achieved with the branches in the data largely preserved. Seurat integration shows a high capacity for removing batch effects, as the cells from multiple days are mixed. However, it is an over-correction. For example, S, S → A and A are mixed, and so do A → A1 and A1, and A → A2 and A2. The same applies to the branch of B. It also leads to incorrect trajectory inferences, as A is now closer to A than A → A1. Overall, it makes it harder to delineate cell differentiation. Seurat with RPCA rather than CCA is slightly less prone to overcorrection but still fails to preserve the A1/2 and B1/2 branches. The result of Harmony is slightly better. The two batches are fairly mixed for A1, A2, B1, and B2. The trajectory of S → A to A, then to A → A1 and A → A2, and finally to A1 and A2 can be seen, although the B → B2 and B2 are misplaced. However, the A branch and B branch are still in disconnected clusters, leaving their relationship with S undiscovered. Overall, LAD performs the best. For time consumption, LAD and Limma use less than 1 second, while Harmony uses 9 seconds and Seurat integration uses 20 seconds.

### Mouse retina development dataset
We applied LAD to a dataset including 13 single-cell specimens collected from the developing retina of one sample at E12 (day 12 embryo), two at E14, one at E16, two at E18, one at P0 (postnatal day 0), two at P2, one at P5, two at P8, and one at P14[5]. A total of 110,359 cells are collected and processed with Seurat, where 2,000 highly variable genes are selected, and 30 PCs are used.

The development of the mouse retina is well-understood by the field (Fig. 3a). Briefly, retinal progenitor cells (RPCs, including early RPCs and late RPCs) differentiate into Neurogenic cells, which further differentiate into photoreceptor precursors, Amacrine cells, horizontal cells, and retinal ganglion cells[14]. The photoreceptor precursors differentiate into cone cells, rod cells, and bipolar cells[15]. These cell types are all neural cells. Late RPCs also differentiate to Müller glia[14]. These cells form a neural network where each terminal cell type has a unique function. Cone cells and rod cells forming the input layer are photoreceptors that work in light and dark environments, respectively. The signal from cone cells and rod cells propagates through the bipolar cells first and then retinal ganglion cells to go to the brain. Horizontal cells provide horizontal connections between photoreceptors and bipolar cells, while amacrine cells perform similar functions between bipolar cells and retinal ganglion cells. These cells form two hidden layers. Müller glia is an auxiliary cell type that supports the aforementioned neural cells.

The result of LAD shows a clear trajectory of the cells gradually evolving from E11 to P14 (Figure 3b), The aforementioned evolution trajectory of the cell types can also be seen along the trajectory. The Euclidean distance also performs relatively well on this dataset. However, a clear batch effect can be seen in uncorrected data (PCA), where cells are clustered by samples. As a result, a cell type is divided into multiple clusters, making it harder to

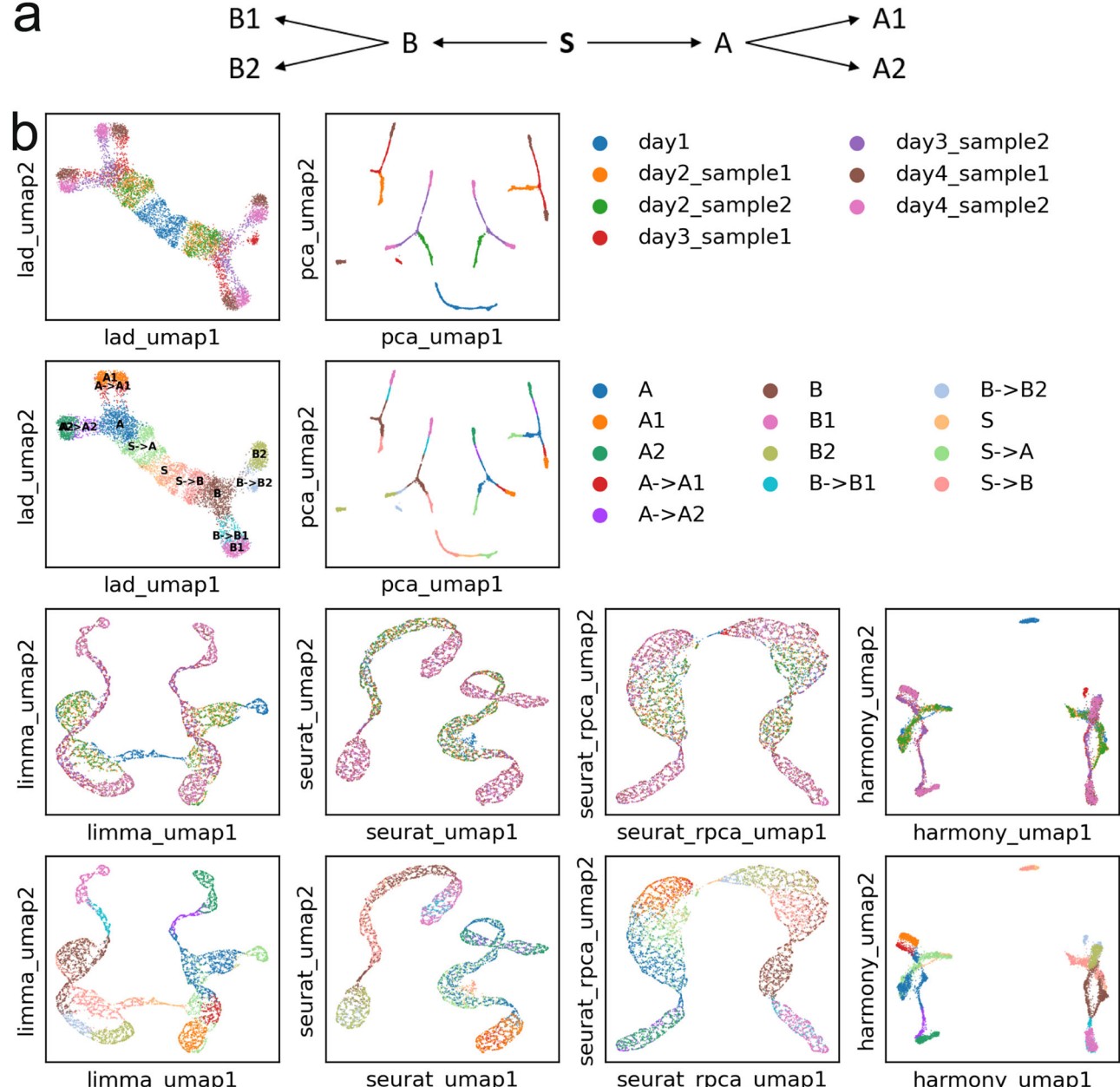

**Fig. 2 | Results on the simulated data. a** Ground truth for the trajectory of cell differentiation. S is the initial state that bifurcates into two branches A and B, and finally into A1, A2, B2, and B. **b** UMAP visualization. Odd numbered rows are colored by samples and even numbered rows are by cell types.

delineate the differentiation trajectory of the cell types. Both Seurat integration and Harmony over-correct the batch effect. Although the samples are well mixed, many cell types are mixed. For example, early RPC, late RPC,

**Table 1 | Simulated dataset**

| Day | Samples | Composition |  |  |  |  |
|---|---|---|---|---|---|---|
|  |  | S | S → A, S → B | A, B | A → A1, A → A2, B → B1, B → B2 | A1, A2, B1, B2 |
| 1 | 1 | 0.5 | 0.5 |  |  |  |
| 2 | 2 |  | 0.5 | 0.5 |  |  |
| 3 | 2 |  |  | 0.5 | 0.5 |  |
| 4 | 2 |  |  |  | 0.5 | 0.5 |

and Müller glia are not distinguishable from the result, and so are horizontal cells, amacrine cells, bipolar cells, and retinal ganglion cells. Even worse, the bipolar cells, which should be differentiated from photoreceptor precursors, are misplaced directly under neurogenic cells. These mistakes are because these methods do not utilize the temporal locality of the samples. Quantitative benchmarks concur with these qualitative evaluations (Figure 3c) LAD shows comparable batch correction strength but conserves more biological content. In addition, trajectory analysis shows that LAD helps Monocle 3 infer pseudotimes more coherent with the actual times and prior biological knowledge about *Nfia*, *Nfib*, and *Nfic* (Supplementary Fig. 2, 3 and Supplementary Note 2)[5]. It should be noted that notwithstanding the better overall embedding, the batch structure is largely retained when zoomed in (Supplementary Fig. 4). Thus, the results may need to be interpreted with caution. For this dataset, LAD uses less than 2 seconds, while Harmony uses 5 minutes. Seurat integration costs more than 3 hours. We also included a

**Article**

similar analysis of developmental data on Zebrafish embryo data in Supplementary Note 3 and Supplementary Fig. 5.

## COVID-19 immune compartment datasets

The global pandemic COVID-19 has reportedly infected 6.3 million people worldwide, with the death toll at 380 thousand. Understanding the immune

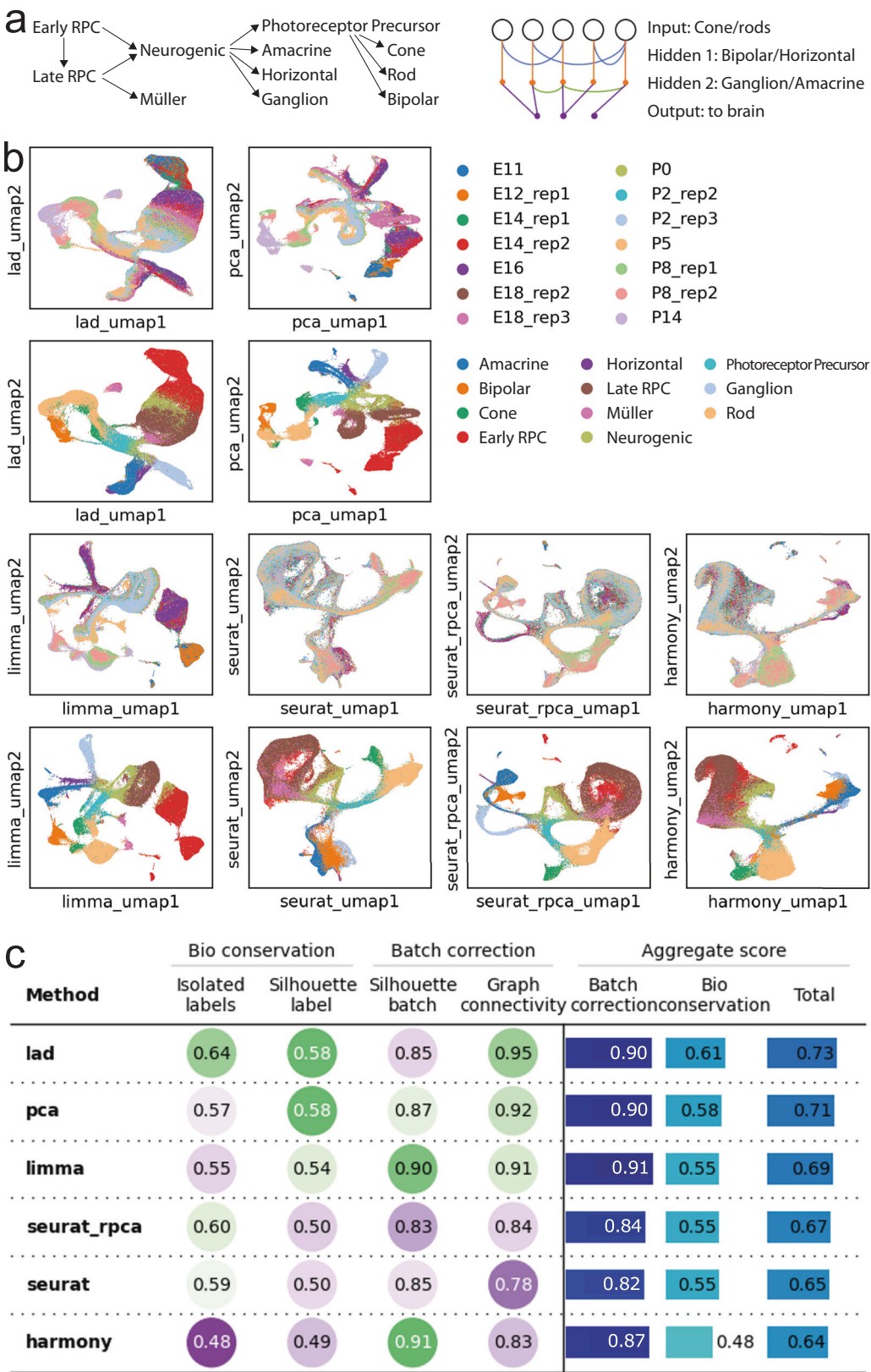

**Fig. 3 | Results for the retina development dataset. a** Trajectory of differentiation of and neural network formed by retinal cells. **b** UMAP visualization of embeddings produced by different methods. The first and third rows are colored by samples and the second and fourth rows are colored by cell types. **c** Quantitative benchmarks of all methods. RPC retinal progenitor cell.

response to the virus is essential to developing treatments. We applied LAD to a recently published dataset including immune compartment samples collected from 13 participants[16] including 4 health controls (HC1-4), 3 moderate (M1-3) cases, and 6 severe (S1-6) cases. Although these are all different participants, we consider HC, M, and S a time course to reflect the progression of the disease. We used Seurat and LAD to process the data. During the process, 2,000 highly variable genes are selected, and 30 PCs are used.

The result is shown in fig. 4a. The largest group is macrophages, which recognize and destroy virus-infected cells. The result shows from top to bottom a gradual change from those in health controls, to moderate cases and severe cases. Some under-correction can be seen in the results. For example, HC3/4 are closer to, but not fully integrated with HC1/2. However, it would still be easier to identify macrophages with the corrected data.

Quantitative benchmarks give similar scores to all methods (Figure 4b), but only LAD delineates the gradual changes in macrophages among health controls, moderate cases, and severe cases (Figure 4a). In contrast, Euclidean distance yields disconnected groups of macrophages because of the batch effect, while Seurat integration (CCA and RPCA), albeit ranked top, and Harmony overcorrect the effect and confuse cells from moderate cases with those from health controls. Limma, on the other hand, appears to be underpowered, leaving many scattered macrophage clusters from different samples. The changes in gene expression that drive the differences can be attained by differential expression analysis. Similar trajectories can also be seen on T cells and plasma cells, which kill infected cells and produce antibodies, respectively. Using these pieces of information, researchers can identify the most effective form of immune cells and find ways to transform others into it to treat the disease.

## Human lung data

Besides studies of the immune compartment, knowledge of lung development may also help cure the disease and restore the functionality of the lung[17]. Miller et al. has produced a single-cell dataset of cells from[18]. Cells from fetal human lungs are collected at week 11.5 (W11.5), W15, W18, and W21, and are available for the trachea, small airways in the lung, and the distal tip of the lung.

We explored the dataset using LAD. Because both temporal and spatial information of the samples is available, we use a vector [*week*, *location*] to label each sample, where *week* is the number of weeks mentioned above, and *location* is set to be 0, 2, and 4 for trachea, small airways, and distal lung, respectively. The result is shown in Fig. 5. A branching trajectory can be seen from W11.5 to W18 for distal lung and small airways mesenchymal cells. The cells from the two locations are similar at the early stage (W11.5) but become more distinct when they are more developed (W15 and W18). In contrast, affected by the batch effect, Euclidean distance and Harmony show W15 and W18 small airway mesenchymal cells as isolated clusters, with no connection with W11.5, while Seurat mixes all mesenchymal cells across the two locations and all times points, blurring the trajectory (Supplementary Fig. 6 and Supplementary Note 4). Similar trajectories also show for Epithelial cells, endothelial cells, and pericytes. Researchers may use these pieces of information to further study the changes in gene expression in the cell type developments and develop treatments.

## Discussion

In order to build a reliable trajectory of cell type development from a longitudinal dataset, the batch effect should be corrected for. This is a new problem as state-of-the-art batch effect correction methods cannot utilize the time/spacial information, and thus result in over-correction. Our method, LAD, utilizes such information to accurately identify and remove the batch effect while preserving the correct structure of the data. The results based on the LAD clearly show the evolution of the cell types through time. The discovered trajectory helps researchers make hypotheses of the physiological changes during development or pathological changes as a result of

disease infection. The changes can be ascertained by differential gene expression analysis.

LAD assumes that the batch effect is shared by all the samples. This is a reasonable assumption because many kinds of batch effects have a biological basis. For example, the cellular stress response stimulated by the sample preparation affects certain genes. In the case the set of genes does change, the local linear approximation may be used. Because the sums and products of distances are guaranteed to be valid distances, multiple LADs each correcting for the batch effect in a specific set of samples can be combined as a consensus distance.

LAD is intrinsically a linear transformation, which may be insufficient for more complex batch effects such as nonlinear interactions of genes in multicenter clinical studies. Notwithstanding the better overall embedding, the batch structure is retained in some cases, necessitating extra caution in interpreting the results." Nevertheless, it is a proof of concept that the time/spatial locality should be considered in batch correction for longitudinal datasets, which are on the rise. Nonlinear methods may be invented based on the same concept. For example, kernelization can be a direct extension to LAD. Researchers usually try multiple quality control, batch correction, and visualization methods to find the most suitable ones for their data, and LAD is an inexpensive option to be included. There is also a potential to cascade LAD with methods like Harmony, which essentially refines the cell-cell similarity graph. Although batch correction helps produce better visualization and clustering results, biases can be introduced during the process. Small false similarity between cells may also be added by the process, although most clustering and visualization methods will ignore them when building the similarity graphs based on only the most similar cells. Overall, rigorous statistical tests on original data are needed to ascertain the findings[19,20].

Although all the experiments are from a biology background, the scope of LAD is not confined to it. It can be applied to any longitudinal/spatial dataset affected by batch effects where the temporal/spatial locality holds. LAD also illustrates that batch the effect correction problem is related to the alternative clustering problems. Over the past few years, many advanced alternative clustering models have been introduced, and translating them to this context may result in better performance.

In summary, we defined a pairwise distance of the cells, namely Label-Aware Distance (LAD), where the effect of the unwanted clustering is controlled. Results show our method achieves more accurate clusters and better visualizations than state-of-the-art methods on longitudinal datasets. The LAD can be directly integrated into most clustering and visualization methods to enable more scientific findings.

## Methods
### The label-aware distance

Trajectory inference, in general, aims to find a graph $G = (V, E)$ which reflects the hop-by-hop gradual change ($E$) of cells ($V_i$, $V_j$, $\cdots \in V$) by optimizing

$$\min \sum_{(V_i, V_j) \in E} \| x_i - x_j \| \tag{1}$$

subjecting to a set of constraints[2,21]. Here, $x_i$ is the profile of cell *i*, which can be the whole gene expression, or the first few principal components (PCs). Euclidean distance is the most widely used metric, while the Mahalanobis distance

$$\| x_i - x_j \|_{\Sigma^{-1}} = \sqrt{(x_i - x_j)^\top \Sigma^{-1} (x_i - x_j)}, \tag{2}$$

where $\Sigma$ is the covariance matrix calculated from all the samples, may be used to account for different (co)variances among features. To account for

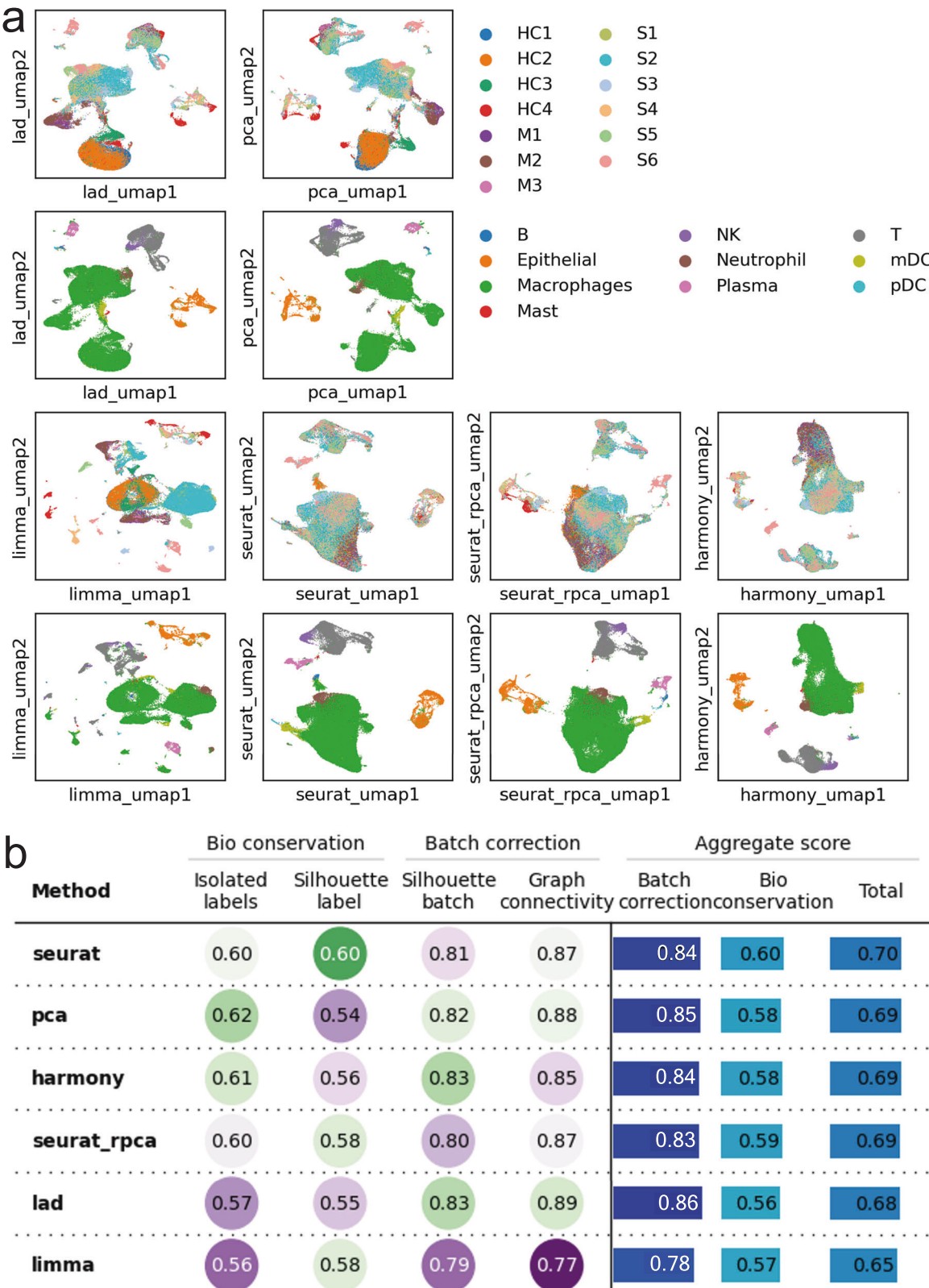

**Fig. 4 | Results on the COVID-19 dataset. a** UMAPs generated from embeddings from all methods. The first and third rows are colored by samples and the second and fourth rows are colored by cell types. **b** Quantitative benchmarks of all methods.

the batch effect, we propose to redefine the $\Sigma$ as

$$\tilde{\Sigma} = \sum_{\text{cell } i=1}^{n} \sum_{\text{batch } b \neq C_i} W_{ib}(x_i - m_b)(x_i - m_b)^{\mathsf{T}}, \quad (3)$$

where $C_i$ is the batch cell $i$ is from, and $m_b$ is the mean expression of cells in batch $b$. It sums over all $b$'s except for the one cell $i$ belongs to. If weight $W_{ib} \equiv 1$, it degrades to the metric defined by Qi and Davidson[22] for generating an alternative clustering. In essence, it removes the variances across the batches, while retaining the variance within each batch. For a

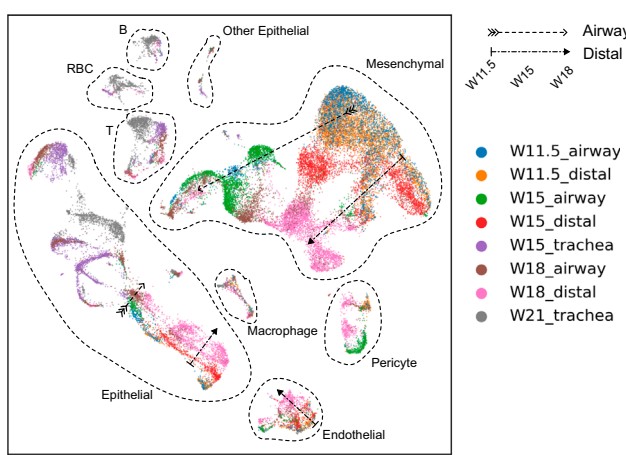

**Fig. 5 | Results on the human fetal lung development dataset.** Arrows are added for visual reference. Figures with detailed cell types are in Supplementary Fig. 6.

longitudinal dataset, each sample $b$ is collected from a time point $t_b$, and a cell $i$ is from the time point $t_{C_i}$. To utilize the temporal/longitudinal locality, we set

$$W_{ib} = \exp\left(-\frac{\| t_{C_i} - t_b \|^2}{2 \, l^2}\right), \qquad (4)$$

where $l$ (set to 1 in our experiments) is the length scale within which two samples are considered temporally/spatially close. The sensitivity to $l$ is generally small (Supplementary Note 5 and Supplementary Fig. 7), and a hyperparameter search can help find suitable ones. The covariance of proximal time points is weighted more and thus is suppressed in the refined distance. When the dataset contains both temporal and spatial labels, $\tau_i$ and $t_j$ can be vectors that include both labels. Inversed Cholesky-decomposed $\tilde{\Sigma}$ can be used to transform the data. If first k PCs are used, the computational complexity is $O(nk^2 + k^3)$. Practically, we solve the linear system $\tilde{\Sigma}^{\frac{1}{2}} y = (x_i - x_j)$ to avoid numerical issues in finding the inverse ($\tilde{\Sigma}^{\frac{1}{2}}$ is the Cholesky decomposition).

### Gene expression data processing

We use Seurat[23], an R package, to analyze the gene expression data. The package provides functionalities to normalize data, find highly variable features (i.e., genes) by variance stabilizing transformation, scale the features, perform principal component analysis (PCA), and visualize the result with UMAP (uniform manifold approximation and projection). This is the de facto standard single-cell data analysis protocol. The normalization step, in particular, normalizes the summation of gene expression in each cell to be one. The scaling step standardizes each gene so that the average expression over all cells is zero, and the standard deviation is one. UMAP is a nonlinear embedding method to visualize data by their distance[24].

Also provided in Seurat is a data integration method that corrects batch effect[9]. It first projects samples into a common subspace using canonical correlation analysis (CCA), and then finds MNNs in the CCA subspace as "anchors" to correct the data. We refer to it as Seurat integration (not to be confused with the entire Seurat protocol). Harmony first projects the data into a lower-dimensional PCA space, and then iteratively removes batch effects. At each iteration, it clusters cells while maximizing the diversity of batches within each cluster and calculates a correction factor for each cell to remove the batch effect. Tran et al.[7] show by systematic assessments that both methods are state-of-the-art. Thus, we compare LAD with them. To ensure good comparability, we implemented LAD with an interface to Seurat. This choice also makes LAD easy to use for biology researchers familiar with Seurat.

Preprocessing is done according to the requirements/recommendations of different methods. Seurat integration handles individually preprocessed data (and uses CCA/RPCA to find a consensus embedding), while other methods working on embedding spaces can only use data that are preprocessed combined. Limma requires log-transformed normalized data before scaling. Because normalization is done per cell, preprocessing separately or combined will give the same result.

### Benchmarks

We added two sets of benchmarks: bio-conservation and batch integration. For bio-conservation, we use silhouette score to measure the overall concordance of cell distribution and ground truth labels and isolated labels silhouette score to measure how well each label is distinguished from all other labels using the average-width silhouette score. For batch integration, we use batch silhouette score to measure how well the batches are mixed and graph connectivity to quantify the connectivity of the subgraph per cell type label. The aggregated score is the arithmetic mean. We used the Scib-metric implementation of these metrics[25].

### Reporting summary

Further information on research design is available in the Nature Portfolio Reporting Summary linked to this article.

### Data availability

All datasets used in this work are public data. They are available in public repositories, including the mouse retina (GSE118614)[26], COVID-19 (GSE145926)[27], and human lung (E-MTAB-8221)[28]. The source data behind Fig. 3c and 4b in the paper can be found in Supplementary Data 1.

### Code availability

The software package is available at https://github.com/KChen-lab/lad and Zenodo[29]. The scripts to generate all results are included in the repository.

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

## Acknowledgements
This publication is part of the Human Cell Atlas - www.humancellatlas.org/publications/. This work was supported in part by grant number 2018-182735 to KC, Human Breast Cell Atlas Seed Network Grant (HCA3-0000000147) to KC from the Chan Zuckerberg Initiative DAF, an advised fund of Silicon Valley Community Foundation, grant RP180248 to KC from Cancer Prevention & Research Institute of Texas, The University of Texas MD Anderson Cancer Center Pre-Cancer Atlas Project to KC, and The University of Texas MD Anderson Cancer Center Colorectal Cancer Moonshot and P30 CA016672 (US National Institutes of Health/National Cancer Institute) to the University of Texas Anderson Cancer Center Core Support Grant.

## Author contributions
S.L. and K.C. conceptualized the problem. S.L., K.C., J.D., and R.I. contributed to the method. S.L. implemented the software and performed the experiments. S.L., K.C., J.D., and R.I. analyzed the results and drafted the manuscript.

## Competing interests
The authors declare no competing interests.

## Ethics statement
The original studies had received sufficient ethical approval from the Institutional Animal Care and Use Committee of the Johns Hopkins University School of Medicine[26], the Research Ethics Committee of Shenzhen Third People's Hospital[27], and The University of Michigan Institutional Review Board[18]. All participants provided written informed consent for sample collection and subsequent analyses[27].

## Statistics and reproducibility
No statistical tests are involved. The computational experiments can be reproduced by using the code provided in the repository.
