## [Peer Review File · Communications Biology]

Reviewers' comments:

Reviewer #1 (Remarks to the Author):

The authors proposed a creative way to leverage the temporal interdependencies among biological samples to enhance the visual clustering and trajectory representation of multi-batch single-cell sequencing data. The demonstrations of the algorithm on the example datasets are very impressive. Nevertheless, I still have several major concerns regarding the contents:

1) The simulated dataset has been intentionally designed to exhibit advantages for the proposed algorithm. For instance, in the case of "day 1 sample 1," a substantial proportion (80%) originates from a single cell state (S). This could make other state-of-the-art algorithms tend towards segregating the data into two major clusters so that they can maximize the distinction between branches A and B. On the contrary, the proposed algorithm can gain an advantage by "peeking" the answer from temporal labels. It compellingly arranges temporally related samples in close proximity so that it can reveal our presumed trajectory patterns. In the context of purpose-driven visualizations, the behavior aligns with the initial design intent, indeed. However, to provide a more comprehensive validation of the algorithm's universality, the authors should consider adding the following tests:

A) Adjust the cell composition of "day 1 sample 1" from (80%, 20%) to (50%, 50%) just like the other samples.

B) In addition to the divergent trajectories, the algorithm should also be tested on convergent trajectories. In fact, I am worried that this algorithm could somehow exaggerate the batch effect by leveraging the information from batches (time labels) so that we "feel" the trajectory fits well with our expectations. This behavior might make sense for divergent trajectory graphs where distant time points are expected to be less biologically alike. But when the real trajectory is not a simple divergent graph, I wonder if this algorithm can still work well.

2) The algorithm is essentially a Euclidean distance-like clustering plus time-label constraints. The clustering results for the real-world datasets are quite similar to the ones obtained from the Euclidean distance method (Fig 3, Fig S1, and Fig S2). In other words, although the time-close samples are arranged proximally to each other, the batch structures are still largely kept, as shown in a) in the simulated dataset, day 4 sample 1 & 2 are still separated (Fig 2b); b) in the COVID-19 dataset, BCD and Euclidean distance generate quite identical results (Fig S1). Maybe we should zoom in and apply a more distinguishable color scheme for samples from different batches of the same condition, respectively.

3) Given that the algorithm was only demonstrated to work well on biological batches (i.e., batches = time points or spatial spots), the name of Batch-Corrected Distance (BCD) is not precise. Maybe the authors should consider a more precise name that explicitly mentions its "label awareness" behavior instead of "batch correction" for its batch "correctness" is yet to be validated.

Minor concerns:

- 1) Supplementary Figures can only be found on GitHub.
- 2) Figure 5 and Figure 2a are similar but not the same.
- 3) The capitalization of journal names should be unified.

Reviewer #2 (Remarks to the Author):

Liang et al. present an interesting approach for correcting inter-batch cell to cell distances in single-cell RNA-seq data sets. The manuscript is well written; however, a few technical points are not clear in the current manuscript. Here are my questions to the authors:

- The authors need to visualize (e.g., show a heatmap of) the batch corrected distance matrix of the simulation data, and compare it to the raw data's as well as the ground truth distance matrices. Seeing what exactly happens to the distance matrix in the simulation could be super helpful in understanding the methods performance on real data.

- In Equation 4, the description of τ_i (for cell i in batch j) is not clear. t_j which is the experimental time point is known, so far so good. But how can one know what is τ_i for each cell?

- With the authors definition of x_i being a cell in batch C_j , and W_{ij} containing information about the cells only in that batch, Equation 3 defines a covariance matrix within each batch C_j . I don't see the inter-batch covariance matrix and distances in this equation, please clarify.

- The authors need to compare results with MNN (Haghverdi et al 2018 <https://doi.org/10.1038/nbt.4091>, package fastMNN <https://rdrr.io/github/LTLA/batchelor/man/fastMNN.html>). Performing a CCA step as Seurat does before matching MNMs, enhances keeping only the shared information among the batches and thus promotes overcorrection. It would be interesting to see if BCD would do better than using the MNN method, without the CCA step.

- In the same publication (Haghverdi et al 2018), it has been shown that linear methods Combat and Limma were not able to correct batch effects in single-cell data, as the authors show single cell batch effects may be only "locally" linear. As covariance correction is basically a linear method, is this type of correction not already taken care of by the previous methods e.g. Limma?

- The application and trial of linear methods for single-cell RNA-seq data batch effect correction has not been further considered after (Haghverdi et al 2018). As BCD is a linear correction method, the authors need to include in their benchmark another linear method such as Combat or Limma.

- In Figure 2 of (Wagner et al. 2018, DOI: 10.1126/science.aar4362), matching mutual nearest

neighboring cells has been used to stitch the longitudinal data from multiple zebrafish embryo developmental time points. The overcorrection of effect between multiple time points seems not too big thus acceptable in this reduced dimension plot. It is interesting to check how BCD correction would compare.

- Whereas BCD sounds plausible for correcting artificial (i.e. batch effect) covariance patterns, subtracting the mean of each batch in any batch effect correction method (as in the operation $x_i - m_j$) sounds risky. Suppose there is a translation difference (e.g. addition of a constant batch effect vector) between two batches. How will the mean of the different batches be placed with respect to each other after the BCD correction?

- In Figure 4, comparison with other methods should also be shown in the main figure, not the supplement, as the result is central for supporting BCDs preference to the other methods.

Reviewer #3 (Remarks to the Author):

Liang and colleagues introduced a novel distance metric termed Batch-Corrected Distance (BCD), designed to mitigate batch effects within single-cell datasets while upholding their underlying longitudinal structure. Addressing batch effects in datasets exhibiting spatial or temporal structures is an important issue. The core concept of the BCD approach involves introducing a weight term to the variance-covariance matrix in the Mahalanobis distance to reflect the temporal/longitudinal locality information. This simple idea yields promising outcomes in both simulated and real-world datasets, although several key methodological and data analysis challenges remain unexplored, rendering the paper more akin to a "proof of concept" (as acknowledged by the authors in the discussion section) rather than a thoroughly assessed and finely polished methodology paper.

Major points:

1. A foundational aspect of this work is the incorporation of a weight term, as defined in equation 4, into the Mahalanobis distance. However, two crucial issues or aspects of arbitrariness arise within this definition. Firstly, the parameter ' l ' can wield a substantial impact on the algorithm's outcomes, as smaller ' l ' values lead to faster decay of cell similarity over time. In extreme cases where ' l ' is significantly smaller than ' $\tau_i - t_i$ ', W_{ij} is close to 0 when i and j are different, indicating interconnections or similarities between cells at different time points could be predominantly disregarded. The sensitivity of outcomes to the choice of ' l ' and strategies for selecting an appropriate ' l ' for real-world datasets warrant exploration.
2. Building upon the first point, the secondary concern tied to the definition of weights pertains to the implicit assumption that average similarities between cells from distinct time points solely hinge on the temporal differences (' $\tau_i - t_i$ ') and adhere to a specific pattern. These assumptions are overly restrictive. For instance, cells from day 2 might differ more substantially from cells on day 1 than from

those on day 3, despite both time intervals being a day long. It's even plausible that cells from day 2 might exhibit greater similarity to cells from day 10 than to those from day 1. In such scenarios, alternative values representing relative differences between time points could offer a better coding solution than using the original day as τ and t , yet the pursuit of such values remains an unresolved query.

3. Notably, the comparisons of different methods on simulated and real data are qualitative rather than quantitative. The inclusion of quantitative metrics is imperative to objectively evaluate the efficacy of distinct methodologies.

4. The paper emphasizes "Clustering and Visualization" in its title, with the abstract asserting that "BCD achieves more accurate clusters and better visualizations than state-of-the-art batch correction methods on longitudinal datasets." While visualization outcomes are qualitatively discussed, clustering results lack both qualitative and quantitative examinations. Moreover, the abstract contends that "BCD can be directly integrated with most clustering and visualization methods to enable more scientific findings," but this integration remains unexplored within the paper. Claims should be supported by concrete evidence/analysis or relegated to the discussion section rather than featuring in the title or abstract.

5. Certain claims require modifications or clarifications. For instance, the statement "the batch effect is uniform across samples" lacks clarity, and substantiating evidence for this assertion is lacking. Similarly, "we randomly selected a set of genes to be 'susceptible to the batch effect'," but the batch effect more or less influences all genes. How to define and select genes 'susceptible to the batch effect'?

6. Is it possible to use trajectory analysis packages such as monocle to help verify/evaluate the comparison in the real datasets?

Minor points:

1. The figure legends necessitate expansion to encompass a detailed interpretation of the figure's implications and the observations derived from it.

2. To facilitate comprehension of computational time comparisons, include additional particulars about the setup. Given that BCD calculates the distance matrix without batch integration, juxtaposing BCD with Seurat and Harmony, which entail comprehensive pipelines for batch effect removal, could require clearer elucidation.

3. Further elaboration is warranted within the methods section. For instance, what algorithm is employed to tackle the optimization problem after including the newly defined weights? Moreover, potential concerns arising from the proximity of the inverse of $\Sigma_{\tilde{}}$ to singularity should be addressed.

4. In Section 4.2, clarify whether multiple datasets are pooled for preprocessing and variable gene selection, or if they are individually processed. In the former scenario, the batch effect could influence preprocessing. If the latter approach is adopted, furnish additional specifics about the procedure.

Response to Comments

We would like to thank all the reviewers for their careful reading and insightful suggestions. We have revised the manuscript accordingly. (Line numbers are referring to the manuscript with version track.)

Reviewers' comments:

Reviewer #1 (Remarks to the Author):

The authors proposed a creative way to leverage the temporal interdependencies among biological samples to enhance the visual clustering and trajectory representation of multi-batch single-cell sequencing data. The demonstrations of the algorithm on the example datasets are very impressive. Nevertheless, I still have several major concerns regarding the contents:

1) The simulated dataset has been intentionally designed to exhibit advantages for the proposed algorithm. For instance, in the case of "day 1 sample 1," a substantial proportion (80%) originates from a single cell state (S). This could make other state-of-the-art algorithms tend towards segregating the data into two major clusters so that they can maximize the distinction between branches A and B. On the contrary, the proposed algorithm can gain an advantage by "peeking" the answer from temporal labels. It compellingly arranges temporally related samples in close proximity so that it can reveal our presumed trajectory patterns. In the context of purpose-driven visualizations, the behavior aligns with the initial design intent, indeed. However, to provide a more comprehensive validation of the algorithm's universality, the authors should consider adding the following tests:

A) Adjust the cell composition of "day 1 sample 1" from (80%, 20%) to (50%, 50%) just like the other samples.

B) In addition to the divergent trajectories, the algorithm should also be tested on convergent trajectories. In fact, I am worried that this algorithm could somehow exaggerate the batch effect by leveraging the information from batches (time labels) so that we "feel" the trajectory fits well with our expectations. This behavior might make sense for divergent trajectory graphs where distant time points are expected to be less biologically alike. But when the real trajectory is not a simple divergent graph, I wonder if this algorithm can still work well.

A) We appreciate the suggestion to perform a more comprehensive simulation study. Figure R1 shows the requested simulation. The result largely resembles the previous one, showing that the previous 80%/20% setting did not favor BCD. To avoid similar concerns from the readers, we replaced the simulation with this result (**Table 1 and Figure 2b**).

Figure R1. Revised simulation results.

B) A convergent trajectory would be a reversed divergent trajectory. Thus, BCD can handle it in the same way as a divergent one, because BCD uses the differences between the time stamps, not the actual values. It is indifferent to an additive constant or a flipped sign, the two operations needed to reverse the time course.

It is a valid concern that leveraging information from batches causes false discoveries. That is why it is recommended that batch-corrected data be used for exploratory analysis and that the findings be rigorously ascertained on the original data with statistical tests (Leucken et al. 2019). Thus, the problem should be largely avoidable. In fact, existing batch correction methods utilize a spectrum of different information, from discrete batches to manually annotated cell types (Leucken et al. 2021). Here, we hope to add to the spectrum a method that considers the meaning of the batches. We now discuss this in **Discussion**.

2) The algorithm is essentially a Euclidean distance-like clustering plus time-label constraints. The clustering results for the real-world datasets are quite similar to the ones obtained from the Euclidean distance method (Fig 3, Fig S1, and Fig S2). In other words, although the time-close samples are arranged proximally to each other, the batch structures are still largely kept, as shown in a) in the simulated dataset, day 4 sample 1 & 2 are still separated (Fig 2b); b) in the COVID-19 dataset, BCD and Euclidean distance generate quite identical results (Fig S1). Maybe we should zoom in and apply a more distinguishable color scheme for samples from different batches of the same condition, respectively. We appreciate the Reviewer's suggestion to investigate the plots more carefully. We redrew the figures with a better color palette to enhance visibility.

3) Given that the algorithm was only demonstrated to work well on biological batches (i.e., batches = time points or spatial spots), the name of Batch-Corrected Distance (BCD) is not precise. Maybe the authors should consider a more precise name that explicitly mentions its "label awareness" behavior instead of "batch correction" for its batch "correctness" is yet to be validated.

We appreciate the suggestion. We propose to change it to "Label-aware distance (LAD)". To avoid confusion, we did not change it in this revision. We will do so after all other concerns are satisfactorily addressed.

Minor concerns:

- 1) Supplementary Figures can only be found on GitHub. We are sorry for the omission. It is included in the revised version, with added contents.
- 2) Figure 5 and Figure 2a are similar but not the same. That is a good catch. The random seed slightly changed the result of UMAP. We have fixed it achieve a more reproducible result and redrawn the figure.
- 3) The capitalization of journal names should be unified. It has been fixed.

Reviewer #2 (Remarks to the Author):

Liang et al. present an interesting approach for correcting inter-batch cell to cell distances in single-cell RNA-seq data sets. The manuscript is well written; however, a few technical points are not clear in the current manuscript. Here are my questions to the authors:

1. The authors need to visualize (e.g., show a heatmap of) the batch corrected distance matrix of the simulation data, and compare it to the raw data's as well as the ground truth distance matrices. Seeing what exactly happens to the distance matrix in the simulation could be super helpful in understanding the methods performance on real data.

We thank the reviewer for the suggestion. The similarity between samples from the same or adjacent time points (e.g., "A2_d4_s1 and A2_d4_s2" and "S and S->A" in Figure R2c) is lost in the noisy version of the data (Figure R2a). The similarity is recaptured by BCD. Some small similarities are added to closer

time points, which may be considered artifacts, but graph-based visualization methods (e.g., UMAP) can suppress this kind of noise.

The result is included in **Supplementary Materials**.

Figure R2. Distance between cells in each cell type of the uncorrected (a) and BCD corrected (b) data and ideal distance between cells before noise was added (c). Each cell type is down-sampled to five cells per sample. The min and max cutoff value for the color bar is determined by 1/3 and 3 times the 0.9 percentile of all distances to emphasize the more similar cells. Shorter distances (darker color) indicate higher similarities. “d” stands for “day” and “s” stands for “sample” (for a specific day).

2. In Equation 4, the description of τ_i (for cell i in batch j) is not clear. t_j which is the experimental time point is known, so far so good. But how can one know what is τ_i for each cell?

For every cell i from sample j , $\tau_i = t_j$. We intended to use t and τ to distinguish cells and samples. To avoid confusion, we have revised our description in **Methods (line 246)**.

3. With the authors definition of x_i being a cell in batch C_j , and W_{ij} containing information about the cells only in that batch, Equation 3 defines a covariance matrix within each batch C_j . I don't see the inter-batch covariance matrix and distances in this equation, please clarify.

The "shifted covariance" is calculated between cells and the mean of the batches it is *not* from. We have revised the description in **Methods (line 241)** for clarity.

4. The authors need to compare results with MNN (Haghverdi et al 2018 <https://doi.org/10.1038/nbt.4091>, package fastMNN <https://rdr.io/github/LTLA/bachelor/man/fastMNN.html>). Performing a CCA step as Seurat does before matching MNNs, enhances keeping only the shared information among the batches and thus promotes overcorrection. It would be interesting to see if BCD would do better than using the MNN method, without the CCA step.

We thank the reviewer for the great insight. We now use the reciprocal PCA pipeline provided in Seurat to alleviate the problem. As stated in their tutorial, it largely follows the requested procedure and thus is less susceptible to overcorrection and applicable to less biologically aligned samples (https://satijalab.org/seurat/articles/integration_rpca). It shows slightly improved performance over CCA on the retina dataset but does not overperform BCD. The results on the COVID-19 data are mixed.

Method	Bio conservation		Batch correction		Aggregate score		
	Isolated labels	Silhouette label	Silhouette batch	Graph connectivity	Batch correction	Bio conservation	Total
bcd	0.64	0.58	0.85	0.95	0.90	0.61	0.73
pca	0.57	0.58	0.87	0.92	0.90	0.58	0.71
limma	0.55	0.54	0.90	0.91	0.91	0.55	0.69
seurat_rpca	0.60	0.50	0.83	0.84	0.84	0.55	0.67
seurat	0.59	0.50	0.85	0.78	0.82	0.55	0.65
harmony	0.48	0.49	0.91	0.83	0.87	0.48	0.64

Figure R3. Results on retina data.

Method	Bio conservation		Batch correction		Aggregate score		
	Isolated labels	Silhouette label	Silhouette batch	Graph connectivity	Batch correction	Bio conservation	Total
seurat	0.60	0.60	0.81	0.87	0.84	0.60	0.70
pca	0.62	0.54	0.82	0.88	0.85	0.58	0.69
harmony	0.61	0.56	0.83	0.85	0.84	0.58	0.69
seurat_rpca	0.60	0.58	0.80	0.87	0.83	0.59	0.69
bcd	0.57	0.55	0.83	0.89	0.86	0.56	0.68
limma	0.56	0.58	0.79	0.77	0.78	0.57	0.65

Figure R4. Results on COVID-19 data.

5. In the same publication (Haghverdi et al 2018), it has been shown that linear methods Combat and Limma were not able to correct batch effects in single-cell data, as the authors show single cell batch effects may be only “locally” linear. As covariance correction is basically a linear method, is this type of correction not already taken care of by the previous methods e.g. Limma?

That is a great point. We explained in **Discussion** that our method likely will lack the power to integrate very complex batches, such as data from multicenter clinical trials. The patterns of batch effect may vary from data to data, and it is common practice for researchers to try a few methods before they decide to use one for further analysis. Our method appears to perform well on data collected from studies with relatively well-controlled environments. A key difference from Limma to allow us to achieve that is that we provided a method specifically for longitudinal data. Comparisons are included in our response to the next question. We revised the **Discussion** section to better address this concern.

6. The application and trial of linear methods for single-cell RNA-seq data batch effect correction has not been further considered after (Haghverdi et al 2018). As BCD is a linear correction method, the authors need to include in their benchmark another linear method such as Combat or Limma.

We appreciate the suggestion and include Limma in the comparison (Figure R1, R3, R4, R5). We observed that BCD generally outperforms or have similar performance as Limma by qualitative and quantitative evaluations. The label-aware design of BCD likely fits the applications better.

7. In Figure 2 of (Wagner et al. 2018, DOI: 10.1126/science.aar4362), matching mutual nearest neighboring cells has been used to stitch the longitudinal data from multiple zebrafish embryo developmental time points. The overcorrection of effect between multiple time points seems not too big thus acceptable in this reduced dimension plot. It is interesting to check how BCD correction would compare.

Thanks for the suggestion. The original publication obtained a good mutual nearest neighbor graph without batch effect correction, suggesting that the batch effect is not large. Quantitative metrics shows that Limma has the top performance while PCA (uncorrected data) and BCD follows (Figure R5). However, visually, all methods other than BCD and PCA mix pluripotent cells with differentiated cells, a sign of overcorrection. BCD fails to integrate 4-10 samples well, but cells from 10-24 samples are more reasonably clustered than other methods.

Method	Bio conservation		Batch correction		Aggregate score		
	Isolated labels	Silhouette label	Silhouette batch	Graph connectivity	Batch correction	Bio conservation	Total
limma	0.56	0.52	0.77	0.85	0.81	0.54	0.65
bcd	0.57	0.51	0.72	0.83	0.77	0.54	0.63
pca	0.57	0.51	0.71	0.83	0.77	0.54	0.63
harmony	0.50	0.48	0.79	0.70	0.74	0.49	0.59
seurat_rpca	0.50	0.49	0.78	0.52	0.65	0.49	0.56

Figure R5. Results on Zebrafish embryo data. (Seurat Integration with CCA failed for running out of memory.)

We noticed that none of these methods visualize the data as well as the original publication. This may be due to differences in preprocessing, such as feature selections. We did not have the resource to reproduce the original result using the provided MATLAB code.

8. Whereas BCD sounds plausible for correcting artificial (i.e. batch effect) covariance patterns, subtracting the mean of each batch in any batch effect correction method (as in the operation $x_i - m_j$) sounds risky. Suppose there is a translation difference (e.g. addition of a constant batch effect vector) between two batches. How will the mean of the different batches be placed with respect to each other after the BCD correction?

That is a great question. Assuming that there are two identical samples that are made different by a constant vector Δ . One group has gene expressions x_i and mean m , and the other has $(x_i + \Delta)$ and $(m + \Delta)$. The shifted covariance matrix will be $\sum(x_i - (m + \Delta))(x_i - (m + \Delta))^T + \sum((x_i + \Delta) -$

$m)(x_i + \Delta) - m)^T = 2 \sum [(x_i - m)(x_i - m)^T + \Delta^2]$. It will not fully remove the translation but will deprioritize the genes that change more in the translation, as illustrated by the simulation study.

9. In Figure 4, comparison with other methods should also be shown in the main figure, not the supplement, as the result is central for supporting BCDs preference to the other methods.

Thanks for the suggestion. They (with additional comparisons) are included in the main figure now. The associated supplementary text is also integrated with the main text.

Reviewer #3 (Remarks to the Author):

Liang and colleagues introduced a novel distance metric termed Batch-Corrected Distance (BCD), designed to mitigate batch effects within single-cell datasets while upholding their underlying longitudinal structure. Addressing batch effects in datasets exhibiting spatial or temporal structures is an important issue. The core concept of the BCD approach involves introducing a weight term to the variance-covariance matrix in the Mahalanobis distance to reflect the temporal/longitudinal locality information. This simple idea yields promising outcomes in both simulated and real-world datasets, although several key methodological and data analysis challenges remain unexplored, rendering the paper more akin to a "proof of concept" (as acknowledged by the authors in the discussion section) rather than a thoroughly assessed and finely polished methodology paper.

Major points:

1. A foundational aspect of this work is the incorporation of a weight term, as defined in equation 4, into the Mahalanobis distance. However, two crucial issues or aspects of arbitrariness arise within this definition. Firstly, the parameter ' l ' can wield a substantial impact on the algorithm's outcomes, as smaller ' l ' values lead to faster decay of cell similarity over time. In extreme cases where ' l ' is significantly smaller than ' $\tau_i - t_j$ ', W_{ij} is close to 0 when i and j are different, indicating interconnections or similarities between cells at different time points could be predominantly disregarded. The sensitivity of outcomes to the choice of ' l ' and strategies for selecting an appropriate ' l ' for real-world datasets warrant exploration.

We agree that this is a valid concern and thank the reviewer for deducing the logic in choosing l for different level of discrepancies among samples. Figure R6 shows on the retina dataset that that the results are stable over a wide range of l (0.1 to 10). However, the sensitivity to the parameter may vary depending on the data. Batch integration as an exploratory data analysis step does require some trial-and-error. We included this message in **Methods**.

Method	Bio conservation		Batch correction		Aggregate score		
	Isolated labels	Silhouette label	Silhouette batch	Graph connectivity	Batch correction	Bio conservation	Total
bcd_1	0.64	0.58	0.85	0.95	0.90	0.61	0.73
bcd_0.5	0.63	0.59	0.85	0.95	0.90	0.61	0.73
bcd_5	0.64	0.58	0.85	0.95	0.90	0.61	0.72
bcd_10	0.64	0.58	0.85	0.95	0.90	0.61	0.72
bcd_0.1	0.63	0.58	0.84	0.95	0.90	0.61	0.72

Figure R6. Sensitivity to hyperparameter l .

2. Building upon the first point, the secondary concern tied to the definition of weights pertains to the implicit assumption that average similarities between cells from distinct time points solely hinge on the temporal differences (' $\tau_i - t_i$ ') and adhere to a specific pattern. These assumptions are overly restrictive. For instance, cells from day 2 might differ more substantially from cells on day 1 than from those on day 3, despite both time intervals being a day long. It's even plausible that cells from day 2 might exhibit greater similarity to cells from day 10 than to those from day 1. In such scenarios, alternative values representing relative differences between time points could offer a better coding solution than using the original day as τ and t , yet the pursuit of such values remains an unresolved query.

We agree that this is a tacit assumption of our method, and the circular trajectory described by the reviewer would violate this assumption. Because our method allows for multidimensional labels, the user can label the batches with points on the unit circle: $(\cos(t/2\pi T), \sin(t/2\pi T))$. We acknowledge that this will require additional knowledge about the biological sample, but this is a common requirement even for trajectory analysis methods (Saelens et al. 2019). Determining the shape of the trajectory is beyond the scope of this study. We hope to provide a simplistic way that works reasonably well on developmental data, as illustrated by the retina samples from a long developmental process.

3. Notably, the comparisons of different methods on simulated and real data are qualitative rather than quantitative. The inclusion of quantitative metrics is imperative to objectively evaluate the efficacy of distinct methodologies.

We agree that quantitative metrics are important. A portion of this work was done before they were widely available. We now added the metrics. **Figures R3** and **R4** show quantitative benchmarks for the retina and COVID-19 datasets. An additional zebrafish developmental dataset is shown in **Figure R5**. Note that some of the metrics are designed for discrete cell types and may not appreciate the trajectories in developmental processes. The quantitative results are added to the main figures and descriptions of these metrics are added to **Methods**.

4. The paper emphasizes "Clustering and Visualization" in its title, with the abstract asserting that "BCD achieves more accurate clusters and better visualizations than state-of-the-art batch correction methods on longitudinal datasets." While visualization outcomes are qualitatively discussed, clustering results lack both qualitative and quantitative examinations. Moreover, the abstract contends that "BCD can be directly integrated with most clustering and visualization methods to enable more scientific findings," but this integration remains unexplored within the paper. Claims should be supported by concrete evidence/analysis or relegated to the discussion section rather than featuring in the title or abstract.

We appreciate the comment, we edited the Abstract to reflect our findings more accurately (Line 14).

In addition, we believe the quantitative metrics shown above addressed clustering performance. For example, the Silhouette scores are widely used to benchmark clustering results. Note that these scores are calculated without running perform clustering to reflect the average performance of clustering. Our experience has been that ARI of actual clustering results tends to be very arbitrary and varies wildly over random seeds. It can support any claims (i.e., not falsifiable).

5. Certain claims require modifications or clarifications. For instance, the statement "the batch effect is uniform across samples" lacks clarity, and substantiating evidence for this assertion is lacking. Similarly, "we randomly selected a set of genes to be 'susceptible to the batch effect'," but the batch effect more or less influences all genes. How to define and select genes 'susceptible to the batch effect'?

We appreciate these suggestions. We have made the following edits to be more accurate:

"The batch effect is uniform across samples, while the biological effect increases over the distance of samples." → "The biological difference increases over the distance of samples, while the batch effect between samples does not."

If I understand it correctly, 'susceptible to the batch effect' refers to our simulation study. The group of genes is randomly selected here. We agree that it is an oversimplified case to illustrate how BCD works and real data can be more complex. We hope the real-world performance of BCD is satisfactorily illustrated by the real-data examples.

6. Is it possible to use trajectory analysis packages such as monocle to help verify/evaluate the comparison in the real datasets?

This is a great suggestion. Here, we used Monocle 3 to infer trajectories to show how the batch effect and different correction methods affect the results (**Supplementary Figure 5**). BCD shows more consistent distribution of pseudotime between repeats at the same time point, and an overall more monotonic increasing trend. The original publication identifies NFI transcription factors (genes: *Nfia*, *Nfib*, and *Nfic*) as candidate regulators of temporal patterning in the developing retina, characterized by a unimodal expression in RPCs over time. BCD faithfully illustrates this trend. By contrast, Seurat and Harmony produced pseudotime that are nearly indistinguishable between E14 and P5, and consequently created erroneous curves for NFI genes. The pseudotime with uncorrected data are acceptable notwithstanding some artifacts for E11, E12, and P14. However, the uncorrected data fail to capture the

decreasing pattern of *Nfia* and *Nfib*, likely due to the artifact in P14. Limma shows oscillating expression of NFI genes, implying that the ordering of cells may be suboptimal. Overall, the label-aware correction strategy can lead to better trajectory/pseudotime inference results.

Figure R7. Trajectory inference with Monocle 2 on retina data. Left column: Distribution of inferred pseudotime within each sample. Right column: Average expression at each pseudotime in RPCs.

Figure R7 (cont'd)

Minor points:

1. The figure legends necessitate expansion to encompass a detailed interpretation of the figure's implications and the observations derived from it.

We revised the legends. More detailed interpretation is included in the main text.

2. To facilitate comprehension of computational time comparisons, include additional particulars about the setup. Given that BCD calculates the distance matrix without batch integration, juxtaposing BCD with Seurat and Harmony, which entail comprehensive pipelines for batch effect removal, could require clearer elucidation.

Thanks for the suggestion. We added it in the **Discussion (line 215)**.

3. Further elaboration is warranted within the methods section. For instance, what algorithm is employed to tackle the optimization problem after including the newly defined weights? Moreover, potential concerns arising from the proximity of the inverse of $\Sigma_{\tilde{}}$ to singularity should be addressed.

Thanks for the suggestion. More details are added in **Methods (line 253)**.

4. In Section 4.2, clarify whether multiple datasets are pooled for preprocessing and variable gene selection, or if they are individually processed. In the former scenario, the batch effect could influence preprocessing. If the latter approach is adopted, furnish additional specifics about the procedure.

It is done according to the requirements/recommendations of different methods. For example, Seurat integration handles individually preprocessed data (and use CCA/RPCA to find a consensus embedding), while other methods working on embedding spaces can only use data that are preprocessed combined. We added these details in **Methods (line 274)**.

Reviewers' comments:

Reviewer #1 (Remarks to the Author):

Most of my questions have been addressed well, except for the major concern #2:

"2) The algorithm is essentially a Euclidean distance-like clustering plus time-label constraints. The clustering results for the real-world datasets are quite similar to the ones obtained from the Euclidean distance method (Fig 3, Fig S1, and Fig S2). In other words, although the time-close samples are arranged proximally to each other, the batch structures are still largely kept, as shown in a) in the simulated dataset, day 4 sample 1 & 2 are still separated (Fig 2b); b) in the COVID-19 dataset, BCD and Euclidean distance generate quite identical results (Fig S1). Maybe we should zoom in and apply a more distinguishable color scheme for samples from different batches of the same condition, respectively."

It turns out the response letter and the revised manuscript did not address my questions above. I expected more explanation than just changing the color palette.

Reviewer #2 (Remarks to the Author):

The authors have addressed my questions. Two minor comments:

- For Supplementary figure 3, they write "...but graph-based visualization methods (e.g., UMAP) can suppress this kind of noise.". The rationale behind this claim is unclear and is worth 1-2 explanatory sentences (e.g. in the Discussion).

-There are two figure numbers as supplementary figure 3.

Reviewer #3 (Remarks to the Author):

The authors have successfully addressed my previous concerns.

Reviewers' comments:

Reviewer #1 (Remarks to the Author):

Most of my questions have been addressed well, except for the major concern #2:

"2) The algorithm is essentially a Euclidean distance-like clustering plus time-label constraints. The clustering results for the real-world datasets are quite similar to the ones obtained from the Euclidean distance method (Fig 3, Fig S1, and Fig S2). In other words, although the time-close samples are arranged proximally to each other, the batch structures are still largely kept, as shown in a) in the simulated dataset, day 4 sample 1 & 2 are still separated (Fig 2b); b) in the COVID-19 dataset, BCD and Euclidean distance generate quite identical results (Fig S1). Maybe we should zoom in and apply a more distinguishable color scheme for samples from different batches of the same condition, respectively." It turns out the response letter and the revised manuscript did not address my questions above. I expected more explanation than just changing the color palette.

Thank you for your comments and sorry for missing the point. We agree that zooming in on investigations can help understand the results. In Figure RR1, we show the detailed alignment of the two replicates of E14. It appears that notwithstanding the better overall embedding, the batch structure is largely retained, as the reviewer correctly pointed out. It appears to depend on the specific cell types. For example, ERPC cells from two replicates are integrated better than RGC, HC, and Photoreceptor progenitors. We have added this statement to the manuscript (Line 147 and Supplementary Figure 7).

Fig. RR1 Zoomed in view of remaining batch effect. ERPC cells from two replicates are integrated better than RGC, HC, and Photoreceptor progenitors.

Simulation: "Batch effect structure can still be seen in A1, A2, B1, and B2, but the downstream visualization is improved as samples are now organized by their time." (Line 94)

COVID-19: "Some undercorrection can be seen in the results. For example, HC3/4 are closer to, but not fully integrated with HC1/2. However, it would still be easier to identify macrophages with the corrected data." (Line 163)

We also added additional discussion: "notwithstanding the better overall embedding, the batch structure is retained in some cases, necessitating extra caution in interpreting the results." (Line 211)

We hope these contents will clarify the ability and limitation of our method.

We would also like to mention that, as promised, the method name has been changed to "Label-aware distance (LAD)" in this version.

Reviewer #2 (Remarks to the Author):

The authors have addressed my questions. Two minor comments:

- For Supplementary figure 3, they write "...but graph-based visualization methods (e.g., UMAP) can suppress this kind of noise.". The rationale behind this claim is unclear and is worth 1-2 explanatory sentences (e.g. in the Discussion).

Thank you for the suggestion. We added additional discussion: "Small false similarity between cells may also be added by the process, although most clustering and visualization methods will ignore them when building the similarity graphs based on only most similar cells." (Line 220)

-There are two figures numbers as supplementary figure 3.

Thank you for catching it. We have corrected the numbering in Supplementary Materials and main text.

Reviewer #3 (Remarks to the Author):

The authors have successfully addressed my previous concerns.

Thank you for your help in improving our manuscript.

REVIEWERS' COMMENTS:

Reviewer #1 (Remarks to the Author):

I am very pleased to see the statements added in this revision, which well clarifies the ability and limitation of the proposed method. I thus have no further concerns.